# Influence of Powder Characteristics on the Microstructure and Mechanical Behaviour of GH4099 Superalloy Fabricated by Electron Beam Melting

Shixing Wang [1,2], Shen Tao [3] and Hui Peng [3,4],*

1 School of Materials Science and Engineering, Beihang University, 37 Xueyuan Road, Beijing 100191, China; wsx161@163.com

2 Surface Engineering Research Institute, Chinese Academy of Agricultural Mechanization Sciences, No. 1 Beishatan Road, Beijing 100191, China

3 Research Institute for Frontier Science, Beihang University, 37 Xueyuan Road, Beijing 100191, China; taoshen@buaa.edu.cn

4 Key Laboratory of High-Temperature Structural Materials & Coatings Technology, Ministry of Industry and Information Technology, Beihang University, 37 Xueyuan Road, Beijing 100191, China

* Correspondence: penghui@buaa.edu.cn; Tel.: +86-(010)-82317117

**Abstract:** A Chinese superalloy, GH4099 (~20 vol.% $\gamma'$ phase), which can operate for long periods of time at temperatures of 1173–1273 K, was fabricated by electron beam melting (EBM). Argon gas atomized (GA) and plasma rotation electrode process (PREP) powders with similar composition and size distribution were used as raw materials for comparison. The microstructure and mechanical properties of both the as-EBMed and post-treated alloy samples were investigated. The results show that the different powder characteristics result in different build temperatures for GA and PREP samples, which are 1253 K and 1373 K, respectively. By increasing the building temperature, the EBM processing window shifts towards a higher scanning speed direction. Microstructure analysis reveals that both as-EBM samples show a similar grain width (measured to be ~200 μm), while the size of $\gamma'$ precipitated in the PREP sample (~90 nm) is larger than that of the GA sample (~130 nm) due to the higher build temperature. Fine spherical $\gamma'$ phase precipitates uniformly after heat treatment (HT). Furthermore, intergranular cracking was observed for the as-fabricated PREP sample as a result of local enrichment of Si at grain boundaries. The cracks were completely eliminated by hot isostatic pressing (HIP) and did not re-open during subsequent heat treatment (HT) of solution treatment and aging. The tensile strength of the PREP sample after HIP and HT is ~920 MPa in the building direction and ~850 MPa in the horizontal direction, comparable with that of the wrought alloy.

**Keywords:** Ni-based superalloys; electron beam melting; additive manufacturing; Argon gas atomized; plasma rotation electrode process; powder characteristics

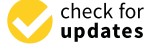



## 1. Introduction

Additive manufacturing (AM) is one of the major driving innovations in different industries. Typically, AM is a powerful tool for the aerospace industry [1–3]. Powder bed fusion (PBF) additive manufacturing, including both selective laser melting (SLM) and electron beam melting (EBM), does not require structural supports, which can optimize the AM building process. In addition, PBF techniques offer higher precision compared with other AM techniques [3]. Recent literature demonstrates that several materials were successfully fabricated by SLM, including 17-4 Ph steel [4], 316 L stainless steel [5], and Ti6Al4V [6,7]. Compared with SLM, the major feature of EBM is the preheated powder bed, which helps alleviate thermal stress during rapid solidification.

Ni-based superalloys are widely used in hot sections in the aerospace industry for their excellent high temperature mechanical properties and oxidation/corrosion resistance [8].

Both SLM and EBM have been utilized for fabricating complex superalloy parts [9,10]. Several previous works have demonstrated the feasibility of fabricating dense and crack free superalloys with a tailored microstructure [11–16]. An ultra-fine crystallographic grain structure was commonly observed in the SLMed superalloys, owing to the rapid solidification rate and high temperature gradients within the micro melt pool. By contrast, coarser columnar grains were usually formed in the EBMed superalloys under preheated conditions [9,17]. Particularly, by carefully controlling the process parameters, a columnar to equiaxed (CET) transition of grains was realized [18–21]. It has also been demonstrated that the mechanical properties of some AM superalloys are comparable to those of wrought and cast superalloys [22,23].

According to the content of Al and Ti, superalloys can be divided into two categories, weldable or nonweldable [12,13,24]. The level of Al + Ti determines the volume fraction of $\gamma'$ precipitates within the superalloy, which plays an important role in the high-temperature strength and creep resistance [23,25,26]. Unfortunately, superalloys with high amount of $\gamma'$ precipitates are considered to be difficult to weld because of their high tendency to crack, particularly as a result of the phenomenon of hot cracking [11,27]. Current SLM and EBM are developed based on welding technologies. However, the processing of nonweldable superalloys brings challenges to AM.

Some recent published works have focused on the cracking mechanisms of AM non-weldable superalloys. It was revealed in Ref. [27] that the hot cracking behavior is attributed to the presence of a liquid film during the last stage of solidification and thermal stresses. A correlation between cracks and high angle grain boundaries (HAGB) was identified. It was also confirmed by atom probe tomography (APT) that a local enrichment of minor element B at the grain boundaries was responsible for the formation of a liquid film at HAGB [27]. Further works performed by Körner [23] and Martin [15] demonstrated the feasibility of producing single crystal superalloy by EBM, in which crack free samples of a nonweldable Ni-based superalloy were successfully obtained by eliminating grain boundaries via competitive grain growth. A very recent work proposed the atomic-grain boundary design [11]. Although these innovative results have aroused widespread interest in the additive manufacturing of superalloys, hot cracks yielded in polycrystalline nonweldable superalloys (columnar or equiaxed grain structure) remain unresolved.

Decreasing the level of $\gamma'$ precipitates in the nonweldable superalloy can reduce the cracking tendency significantly. However, a proper amount of $\gamma'$ precipitates is desirable for adequate mechanical strength. For certain high-temperature applications, the amount of $\gamma'$ precipitates should be optimized to achieve the best balance of strength and weldability. An example of this is the relatively newly designed $\gamma'$-strengthened superalloy, HAYNES 282, which was developed for applications in both aero and land-based gas turbine engines [28]. The total $\gamma'$ forming elements (Al + Ti) content of HAYNES 282 is 3.7 wt.%, with the $\gamma'$ mole fraction of ~19%. The carefully selected $\gamma'$ precipitates level ensures the alloy has a combination of satisfied mechanical properties and improved weldability, compared with the typical non weldable superalloy.

In this work, a Chinese superalloy called GH4099 was fabricated by electron beam melting. This alloy has a similar $\gamma'$ precipitates level with HAYNES 282, and can operate for a long term at temperatures of 1173–1273 K. The objective of the following investigation is to exhibit the influence of raw powder characteristics on the microstructure of the EBMed superalloy. The mechanism of minor element Si on the cracking behavior was elucidated. The evolution of cracks during heat treatment and mechanical properties of the samples were also of concern.

## 2. Experiment

### 2.1. GH4099 Powders

Argon gas atomized (GA) and plasma rotating electrode process (PREP) GH4099 powders were used as raw material, which were provided by CASIC-Hunan and CISRI-Gaona, China, respectively. The chemical composition of both GA and PREP powders

were measured by inductively coupled plasma atomic emission spectroscopy (ICP-OES), as shown in Table 1. The nominal composition of GH4099 specified by *China Aeronautical Materials Handbook* is also listed in Table 2 for comparison. It can be seen that the two powders were provided with similar composition, except for the fact that the Si content in the PREP powders is almost one order of magnitude higher than that of the GA powder. Note that the 0.285 wt.% of Si is quite close to the permissible limit of the composition standard. Powder morphologies are shown in Figure 1. Both of the powders are spherical. However, the presence of small satellites and micro particles can be observed for the GA powders, compared with the perfectly smooth PREP powders. Figure 2 exhibits similar size distribution curves for the two powders measured by a laser granulometry (Bettersize, 9300ST). The powder size ranges primarily between 45 μm and 140 μm, with an average value of ~90 μm. The tap density of two powders was determined to be 4.53 g/cm$^3$ and 4.89 g/cm$^3$, respectively. Flow times of 16.4 s and 15.4 s were measured by a Hall flowmeter (50 g, 2.54 mm orifice), see Table 3.

**Table 1.** Chemical composition of GA and PREP powders of GH4099 (wt.%).

| Powder Type | Element Content, wt.% | | | | | | | |
|---|---|---|---|---|---|---|---|---|
| | **Cr** | **Co** | **W** | **Mo** | **Ti** | **Al** | **Fe** | **Mn** |
| Argon Gas Atomized (GA) | 18.21 | 6.44 | 5.88 | 4.00 | 1.29 | 2.03 | <0.1 | <0.005 |
| | Si | C | N | O | Ni | | | |
| | 0.025 | <0.1 | 0.005 | 0.008 | Bal. | | | |
| Plasma Rotating Electrode Process (PREP) | Cr | Co | W | Mo | Ti | Al | Fe | Mn |
| | 18.65 | 5.91 | 5.91 | 3.89 | 1.29 | 2.26 | 0.54 | 0.24 |
| | Si | C | N | O | Ni | | | |
| | 0.285 | <0.1 | 0.003 | 0.008 | Bal. | | | |

**Table 2.** Nominal composition of GH4099 (wt.%).

| Element | Ni | Mo | Fe | Cr | W | Al | Ti |
|---|---|---|---|---|---|---|---|
| Content | Bal. | 3.50–5.00 | ≤3.00 | 17.50–19.50 | 5.50–7.00 | 2.50–3.00 | 1.00–1.50 |
| Element | Co | Si | B | Mn | Cu | Ce | S/P |
| Content | 5.00–8.00 | ≤0.30 | ≤0.005 | ≤0.30 | ≤0.070 | ≤0.020 | ≤0.015 |

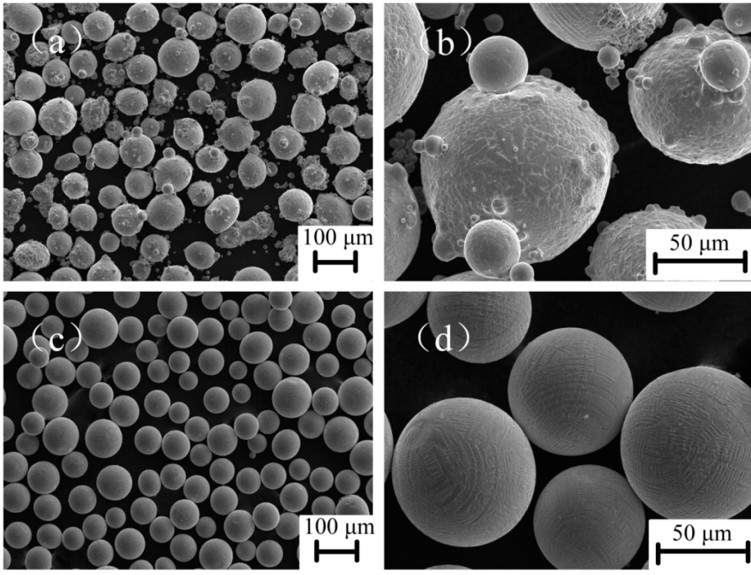

**Figure 1.** SEM morphology of pre-alloyed GH4099: (**a**,**b**) GA; (**c**,**d**) PREP.

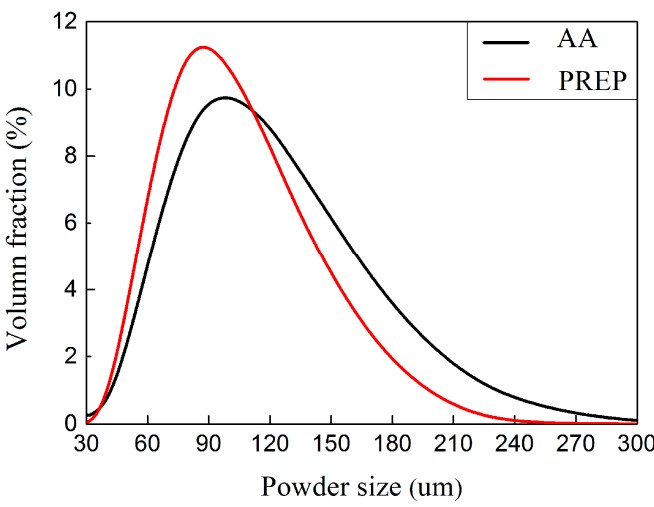

**Figure 2.** Volume fraction distribution of powder particles.

**Table 3.** Powder properties of two types of GH4099.

| Powder | SSA (m$^{-2}$/kg) | Flowability (s) | Packing Density (g/cm$^3$) | D10 (μm) | D50 (μm) | D90 (μm) |
|---|---|---|---|---|---|---|
| GA | 32.91 | 16.4 | 4.53 | 49.68 | 92.57 | 155.0 |
| PREP | 28.36 | 15.4 | 4.89 | 52.74 | 84.00 | 133.8 |

*2.2. EBM Sample Fabrication*

GH4099 superalloy samples were fabricated using an Arcam A2XX machine (Control Software 3.2, Arcam, Stockholm, Sweden) in a manual mode. Figure 3a shows the schematic diagram of one batch of as-EBM samples. Layers were scanned using a snake-like melting strategy (i.e., either 0° or 90° hatch direction was used for each layer), as shown in Figure 3b. A 10 mm thick stainless steel plate with a diameter of 150 mm was used as a starting plate. No supports were applied for the EBM fabrication. One batch of the EBM fabrication included 9 with each 20 value × 20 value × 30 value mm in X-Y-Z dimension. EBM process windows were first established for the GA powders by using a range of scanning speed $v$ from 333–9000 mm/s, combined with a range of line offset $L_{off}$ from 0.1–0.3 mm, followed by the PREP powders. Three different beam current $I$ values of 5 mA, 10 mA, and 15 mA were selected. This resulted in equivalent area energy $E_A$ values ranging from 0.414 J/mm$^2$ to 3 J/mm$^2$. This resulted in equivalent area energy $E_A$ ($E_A = UI/(v \cdot L_{off})$, where $U$ is the operating voltage of 60 kV) values ranging from 0.414 J/mm$^2$ to 3 J/mm$^2$. A similar EBM process window diagram was also established for the PREP powders. Detailed process parameters are summarized in Table 4.

**Table 4.** Summary of the processing parameters of two types of GH4099 powders.

| GA Powder | | | | PREP Powder | | | |
|---|---|---|---|---|---|---|---|
| $I$ (mA) | $v$ (mm/s) | $L_{off}$ (mm) | $E_A$ (J/mm$^2$) | $I$ (mA) | $v$ (mm/s) | $L_{off}$ (mm) | $E_A$ (J/mm$^2$) |
| 5 | 333 | 0.3 | 3.00 | 5 | 3250 | 0.1 | 0.92 |
| 5 | 500 | 0.2 | 3.00 | 5 | 4250 | 0.1 | 0.71 |
| 5 | 833 | 0.3 | 1.20 | 5 | 3750 | 0.1 | 0.80 |
| 5 | 1000 | 0.1 | 3.00 | 10 | 417 | 0.3 | 4.80 |
| 5 | 1666 | 0.15 | 1.20 | 10 | 625 | 0.2 | 4.80 |
| 5 | 3000 | 0.1 | 1.00 | 10 | 833 | 0.3 | 2.40 |
| 5 | 4500 | 0.1 | 0.67 | 10 | 1250 | 0.1 | 4.80 |

**Table 4.** *Cont.*

| GA Powder | | | | PREP Powder | | | |
|---|---|---|---|---|---|---|---|
| $I$ (mA) | $v$ (mm/s) | $L_{off}$ (mm) | $E_A$ (J/mm$^2$) | $I$ (mA) | $v$ (mm/s) | $L_{off}$ (mm) | $E_A$ (J/mm$^2$) |
| 10 | 833 | 0.3 | 2.40 | 10 | 1250 | 0.2 | 2.40 |
| 10 | 1250 | 0.2 | 2.40 | 10 | 1250 | 0.3 | 1.60 |
| 10 | 1250 | 0.3 | 1.60 | 10 | 1875 | 0.2 | 1.60 |
| 10 | 2000 | 0.1 | 3.00 | 10 | 2250 | 0.1 | 2.67 |
| 10 | 2500 | 0.1 | 2.40 | 10 | 2500 | 0.1 | 2.40 |
| 10 | 3000 | 0.1 | 2.00 | 10 | 3750 | 0.1 | 1.60 |
| 10 | 3500 | 0.1 | 1.71 | 10 | 4500 | 0.1 | 1.33 |
| 10 | 4000 | 0.1 | 1.50 | 10 | 6500 | 0.1 | 0.92 |
| 10 | 4500 | 0.1 | 1.33 | 10 | 8500 | 0.1 | 0.71 |
| 10 | 5000 | 0.1 | 1.20 | 10 | 9750 | 0.1 | 0.62 |
| 10 | 5500 | 0.1 | 1.09 | 10 | 10,500 | 0.1 | 0.57 |
| 10 | 6000 | 0.1 | 1.00 | 10 | 12,500 | 0.1 | 0.48 |
| 10 | 6500 | 0.1 | 0.92 | 10 | 14,500 | 0.1 | 0.41 |
| 15 | 1875 | 0.2 | 2.40 | 10 | 16,500 | 0.1 | 0.36 |
| 15 | 1500 | 0.1 | 6.00 | 10 | 18,500 | 0.1 | 0.32 |
| 15 | 2000 | 0.1 | 4.50 | 15 | 6750 | 0.1 | 1.33 |
| 15 | 2500 | 0.1 | 3.60 | 15 | 9750 | 0.1 | 0.92 |
| 15 | 3000 | 0.1 | 3.00 | 15 | 12,750 | 0.1 | 0.71 |
| 15 | 7500 | 0.1 | 1.20 | 15 | 2500 | 0.1 | 3.60 |
| 15 | 9000 | 0.1 | 1.00 | | | | |
| 15 | 9000 | 0.1 | 1.00 | | | | |

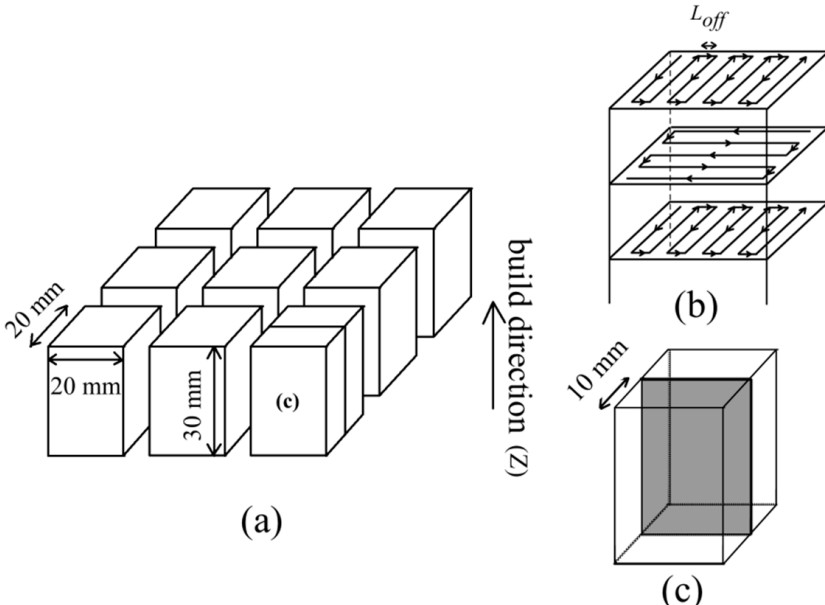

**Figure 3.** Schematic diagrams of (**a**) one batch of as-EBM samples; (**b**) electron beam scans in a cross snake-like way with a specific $L_{off}$ and hatching layer of 50 μm; (**c**) shows the microstructural characterization region.

The build temperature for the GA and PREP powders was determined to be 1253 K and 1373 K, respectively, at which the powder bed was slightly sintered. The monitored temperature was almost constant for the whole EBM fabrication process. The samples were cooled slowly to room temperature after building.

### 2.3. Microstructural Characterisation

As-EBM samples were sectioned along the vertical plane, as illustrated in Figure 3c. Standard metallographic procedures including grinding and polishing were performed. A Gemini SEM 300 (Zeiss, Oberkochen, Germany) was used for scanning electron microscopy (SEM) for microstructural observation. Chemical etchant for revealing the grain boundaries was a solution of 4 g $CuSO_4$, 100 mL HCl, and 100 mL $H_2O$. The polished cross-sections were electro-etched at 5 V for 6 s in a solution of 12 mL $H_3PO_4$, 40 mL $HNO_3$, and 48 mL $H_2SO_4$ to examine the $\gamma'$ precipitations. An electron-probe microanalyzer (EPMA) JXA-8100 (JEOL, Tokyo, Japan) was also employed to detect the minor element concentration at the vicinity of a crack. One thing that should be mentioned is that in order to prevent element contamination during sample preparation, the EPMA samples were cut by diamond wire, ground by diamond abrasive papers, and polished by diamond slurry.

### 2.4. Post-Processing

The samples with internal cracks were healed by hot isostatic pressing (HIP, Model QIH9, Quintus) at 1423 K, 150 MPa for 4 h, referred to as EBM+HIP condition. Selected EBM+HIP samples were then subjected to a solution and aging heat treatment (HT). The standard two-step treatment included 1453 K/2 h/AC (air cool) +1173 K/4 h/AC. This condition is referred to as EBM+HIP+HT. The HT was performed in a horizontal tube furnace in an air atmosphere. Throughout the HT process, a thermocouple was placed next to the samples to monitor the temperature. It is worth noting that the purpose of solution treatment is to make unevenly distributed $\gamma'$ precipitations further re-dissolved into the $\gamma'$ matrix, i.e., the uniform precipitation of $\gamma'$ phases, and to readjust the morphology of $\gamma'$ phase by changing aging temperature, aging time, or cooling method. This is done to realize precipitation strengthening and improve the mechanical properties of the Ni-based superalloy.

### 2.5. Mechanical Property Characterization

Micro-hardness measurements were performed using an FM-800 micro-hardness tester (FUTURE-TECH, Tokyo, Japan). A 200 g load force with a holding time of 10 s was used. Fifteen measurement points were made per sample on the sectioned plane (Figure 3c). The average hardness value is reported together with the standard deviation (STDEV).

Flat tensile specimens were machined with the load axes parallel and perpendicular to the build direction. Specimens in both as-EBM and post-processed conditions were prepared. Tensile tests were performed at room temperature using an electronic universal testing machine CMT4000 (SANS, Shenzhen, China), where strain was determined with an extensometer. A constant cross-head speed of 0.015 mm/s was used.

## 3. Results

### 3.1. Characteristics of Starting Powders for EBM Process

Gas atomization is completed by molten metal which is atomized due to inert gas jets into fine metal droplets, then cooled down in the atomizing tower. The plasma-rotating electrode process is a centrifugal atomization process in a vacuum atmosphere developed by Starmet. Plasma arc is involved in this process, and this method is currently a leading candidate for high-purity powders of metals with a high melting point such as Ti and Ni alloy powder production [29–31].

Representative micrographs (SEM) of atomized powders are shown in Figure 1a,b for GA powders and Figure 1c,d for PREP powders. In general, PREP powders shown in Figure 1c exhibited a much better spherical shape compared to GA powders, Figure 1a.

In addition, small satellite particles can be found in PREP powders, Figure 1c,d. As described in [32], powder particle size distribution and its mean size have traditionally been considered to be important. This has a certain impact on packing density on the build plate as well as the density of the finished part. Powders with smaller average particles (D50) have been found to result in better surface roughness of the completed samples according to [33]. Large particles might not be fully melted and fused, conversely too-small particles can lead to ball, spatter, or swell causing too much energy [34]. However, there are many particle properties that could potentially influence the overall behavior of the powder for the subsequent EBM process. These key particle properties include specific surface area, flowability, packing density, and D90.

Figure 4 shows the interior morphology of GH4099 powder particles. It was clear that the interior of GA powder particles were massive cellular dendrites (Figure 4a,b) due to rapid solidification during the cooling process. The interior morphology of PREPed GH4099 powder particles were mainly segregated dendritic and light from cellular dendrites (Figure 4c,d), resulting from different cooling rates between metal, liquid, and matrix in the solidification process [31], which made the dendrites grow and become coarse.

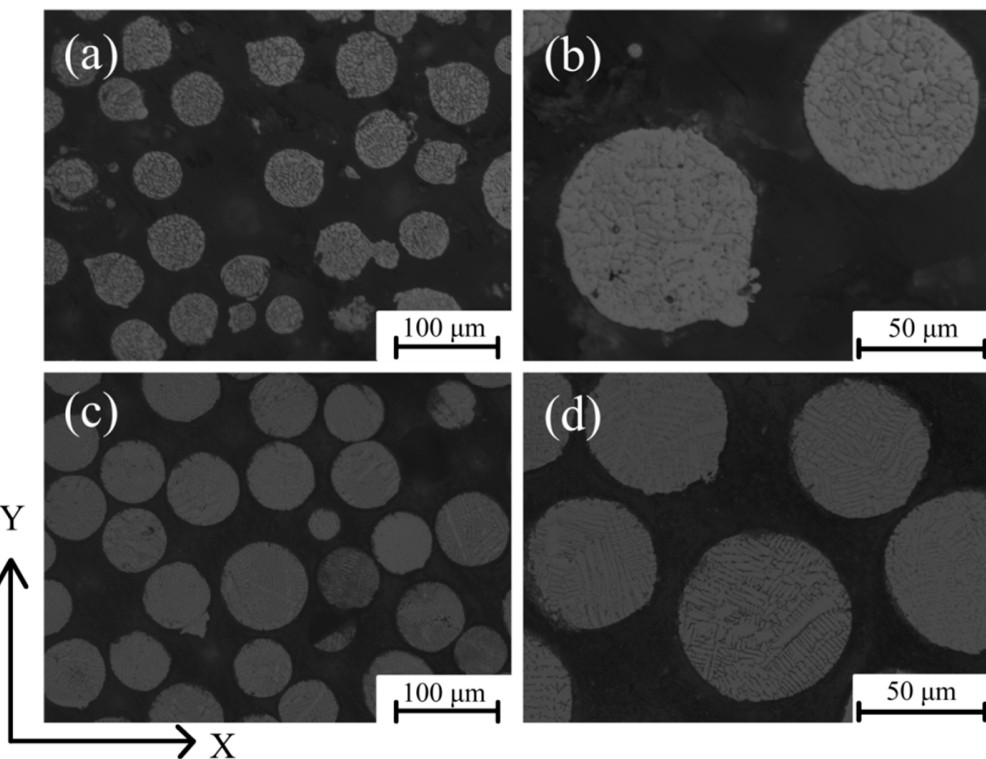

**Figure 4.** SEM micrographs of cross-section of the interior crystalline structure in the particles (**a**,**b**) GA; (**c**,**d**) PREP.

*3.2. Process Window*

According to the results of optical microscope (OM) and SEM, these samples have two types of defects. Namely uneven and bonding defects in the micromorphology occurred during the EBM process window based on the varying melting current and scanning speed. There were also dense samples with no surface unevenness, which were divided into two forming windows according to the different powders used in this work, as shown in Figure 5.

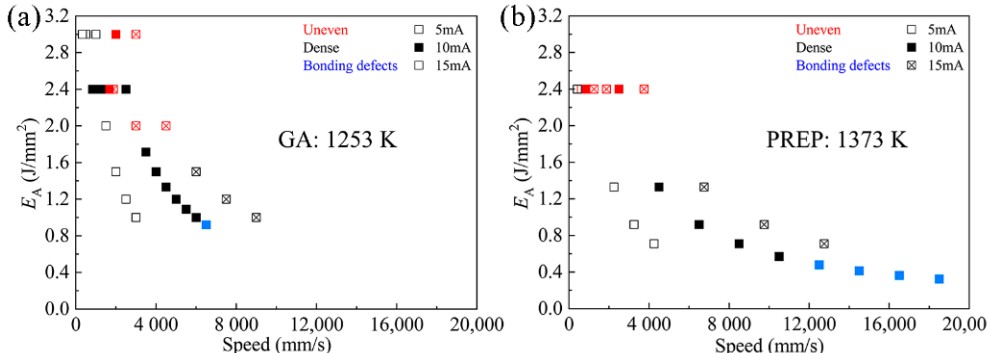

**Figure 5.** EBM processing windows for (**a**) GA powder and (**b**) PREP powder are plotted with scanning speed (*v*) against $E_A$. Suitable parameters (combinations of beam current, scanning speed, and $E_A$) for dense samples with no top surface unevenness and binding defects are indicated by black symbols.

From Figure 5a, the GA powders with the build temperature of 1253 K, unevenness or bonding defects occur when $E_A$ is greater than ~2.0 J·m$^{-2}$ or less than ~1.0 J·m$^{-2}$. For the PREP powders with the build temperature of 1373 K, dense samples could be obtained when $E_A$ is in the range of ~0.5 J·m$^{-2}$ to ~1.5 J·m$^{-2}$ (Figure 5b). It can be concluded for both cases that a suitable combinations of parameters (beam current, scanning speed, and $E_A$) are necessary for defect free samples. By increasing the build temperature, the EBM processing window shifts towards higher speed and lower energy exposure per area $E_A$ (in J·m$^{-2}$) direction. Due to the different powder characteristics, the PREP powders are more difficult to sinter during the preheating of the powder bed by electron beam. This resulted in a higher build temperature for the PREP powders than the GA powders.

Bonding defects occur with increased scanning speed when the energy input is too low to completely melt the powder layer, leaving the longitudinal pores [14]. Unevenness on the surface occurs with lower deflection speeds and a higher scanning current of the electron beam. Helmer et al. [35] reported that unevenness of the top surface is a result of a vigorous melt pool motion caused by high forces exerted on the melt pool which could be interpreted as temperature gradients, surface energy effects (Marangoni convection), and evaporation.

### 3.3. Microstructural Characterization

Columnar crystals and the equiaxial crystals were observed in as-EBM condition samples deposited with GA powder and PREP powder. An obvious columnar to equiaxed transition (CET) was also observed for both powders within the processing window, which is not shown here.

Two samples made from GA and PREP powders were selected for detailed investigations. As can be seen in Figure 6a,b, both samples reveal a similar columnar grain width of ~200 μm. The average grain width of the EBM sample deposited with GA powder (GA sample), as well as the sample as-EBM condition deposited with PREP powder (PREP sample) are 195.16 ± 18.46 μm and 206.47 ± 28.74 μm, respectively. SEM images of the $\gamma'$ precipitations are shown in Figure 6c,d. The distribution of the $\gamma'$ is quite homogeneous in the as-EBM state, with a volume fraction of about 20% for both samples. The $\gamma'$ is seen to be spherical and very fine in size. Differing from the limited formation of $\gamma'$ in the as-SLMed superalloys [36], the processing conditions (rapid solidification and high build temperature) of EBM fabrication may provide a proper thermal history for the precipitation of $\gamma'$. It is therefore not surprising that the higher build temperature resulted in a larger $\gamma'$ precipitated (~90 nm) in the PREP sample than that of the GA sample (~130 nm).

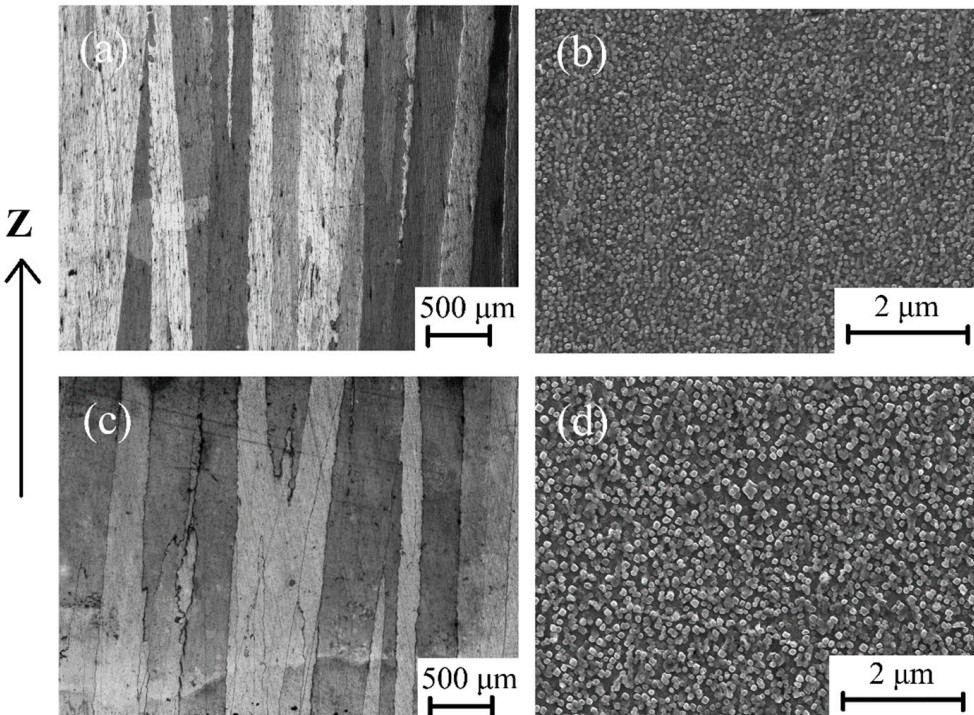

**Figure 6.** Optical micrographs in the X-Z plane of GH4099 samples fabricated by EBM deposited with: (**a**) GA; (**c**) PREP powder, and SEM micrographs of γ′ phase as-EBM GH4099 samples showing the morphology of γ′ precipitates: (**b**) GA; (**d**)PREP.

*3.4. Cracking in as-EBM GH4099 Superalloy*

Macro cracks were not observed on the external surface and the wire-cut surface of the as-EBM GH4099 samples. However, the micro-cracks existed in the samples deposited with PREP powder by optical microscope and scanning electron microscope after grinding and polishing, but not in the samples deposited with GA powder. The cracks are parallel to the building direction, and no transverse cracks or lack of fusion appear, Figure 7. These microcracks are distributed along the columnar grain boundaries, which indicates that the microcracks are intergranular cracks.

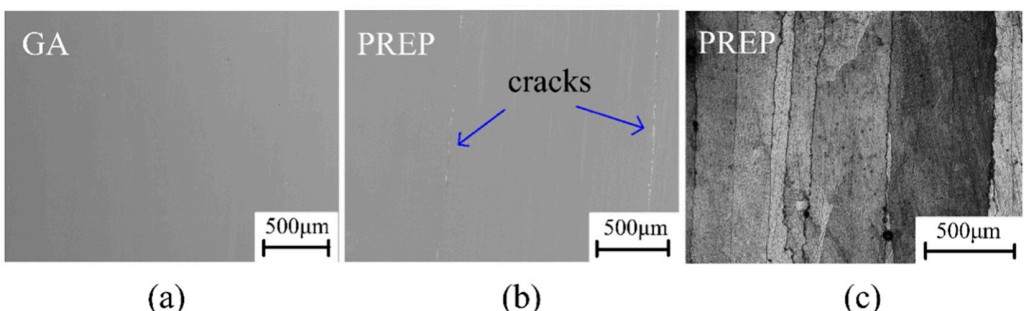

**Figure 7.** Crack morphology of as-EBM GH4099: optical micrographs of un-etched (**a**) GA sample with no cracks and (**b**) PREP sample with cracks parallel to the building direction. (**c**) Etched PREP sample showing the microcracks are intergranular cracks.

Observation of the polished cross-section are shown in Figure 7a,b. It can be seen that the GA sample is dense and crack-free. However, micro-cracks were detected in the PREP sample, with the cracking ratio determined to be ~5% in area fraction. The occurrence of cracks is associated with the formation of liquid films at grain boundaries, as

indicated by the arrows shown in Figure 7b. Figure 7c also confirms that the microcracks are intergranular cracks.

### 3.5. Crack Healing and Microstructural Evolution

Kirka et al. [37] found that holes in Inconel 718 alloy formed by EBM could be removed by hot isostatic pressing. Carter [38] et al. found that cracks in CM247LC alloy formed by SLM could be eliminated by hot isostatic pressing. Therefore, in order to eliminate the micro-cracks in as-EBM sample, selected PREP sample was given a HIP treatment. The SEM micrograph of the GH4099 after HIP indicates that HIP can eliminate the micro-cracks in as-EBMed GH4099. Figure 8a,b shows the microstructure evolution of $\gamma'$ phase and the coarse irregular-shaped $\gamma'$ phase after HIP appeared with the HIP temperature close to the initial melting point of GH4099, between 1423 K and 1453 K, which may affect the microstructure of the sample. Due to the fact that the hatching temperature of 1423 K is much higher than 1223 K in the process window, while the hatching time of a batch of samples is about 30 h, the achieved part retained its heat for a long time, which was equivalent to a high temperature solution treatment of the sample. After a long period of heat preservation, $\gamma'$ phases were sufficiently precipitated and matured.

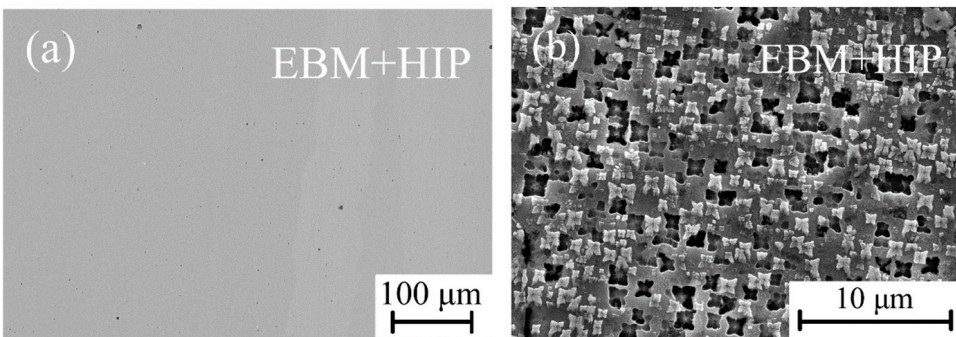

**Figure 8.** SEM micrographs of EBM+HIP PREP sample. Healed cracks after HIP (**a**) and the butterfly-like $\gamma'$ precipitates (**b**).

Through various numerical simulations, Cha et al. [39] found that the growth rate in the <110> direction is enhanced due to a high chemical driving force, while the migration rate in the <100> direction is suppressed due to enriched solute and a corresponding low chemical driving force. The growing shape then transforms into a concave shape. The splitting occurs only in the concave growth condition and it is induced by interface instability, which is in good agreement with experimental observations. Yoo et al. [40] found that morphological unstable and irregular growth in Ni base superalloy occurs whenever there is enough supersaturation in the matrix so that the point effect of diffusion is operative with the change of morphology.

The resultant post-processed combined HIP with HT EBM GH4099 microstructure is illustrated in Figure 9. Heat treatment has a pronounced effect on the structure related to microfissuring through the effect on solute solubility, grain size, and/or the dissolution of phases [41]. Cracks were successfully closed and did not reopen after heat treatment (Figure 9a) which is different from our previous report on a nonweldable superalloy [13]. Figure 9b shows the features of $\gamma'$ precipitates for heat-treated samples. The coarse irregular-shaped $\gamma'$ was optimized into fine spherical particles and distributed more homogeneously in comparison with the HIP sample after heat treatment, which plays an effective role in adjusting the microstructure of the GH4099 alloy. The HIP+HT resulted in grain refinement, while $\gamma'$ precipitates with more volume fraction. This was found after the HT when compared to that of the as-EBM material.

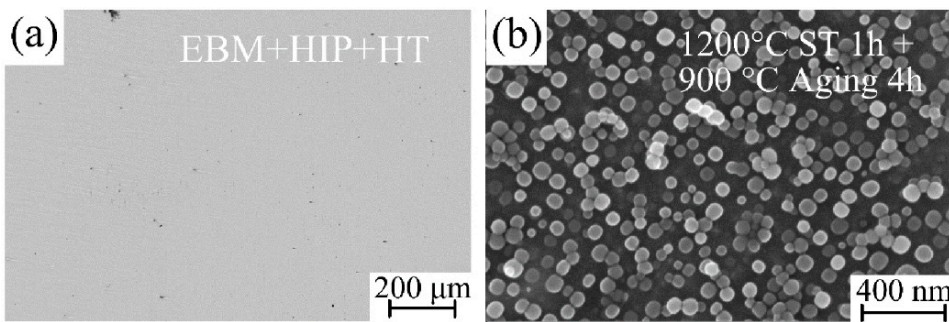

**Figure 9.** (**a**) The cracking incident after HIP+HT and (**b**) microstructural evolution in terms of $\gamma'$ precipitates. The internal structure appears to be crack-free in EBM+HIP+HT condition.

Table 5 summarizes the grain width and $\gamma'$ precipitates volume fraction of the GH4099 superalloy in different conditions. The results show that the difference of $\gamma'$ volume fractions between GA and PREP powder-deposited GH4099 alloys with similar columnar crystal grain widths is small. When passed through HIP treatment of the PREPed sample, the grain width reduced to $70.74 \pm 7.70$ µm. The HIP temperature is 1423 K, which is close to the incipient melting temperature of the GH4099 alloy between 1423 K and 1453 K, thus affecting the microstructure of the sample. The 4 h heat preservation time during the HIP process is far less than the 30 h required for EBM formation of a batch of samples, and the grains had insufficient time to grow. When undergoing HT, the average grain width in the EBM+HIP condition of PREPed sample did not change significantly, but the number of $\gamma'$ precipitates increased. These results show that PREPed sample undergoing HIP treatment then being exposed to 1473 K for 1 h + 1173 K for 4 h HT can not only completely eliminate the coarse irregular-shaped $\gamma'$ precipitates, but also improve the mechanical properties of the alloy without affecting the grain structure of the superalloy.

**Table 5.** Measured average grain width and $\gamma'$ volume fraction of as-EBM GAed and PREPed GH4099 samples and their changes due to HIP and HT. The building parameters of the GAed and PREPed sample are listed here.

| Sample | $I$ (mA) | $V$ (mm/s) | $L_{off}$ (mm) | Grain Width (µm) | $\gamma'$ Volume Fraction (%) |
|---|---|---|---|---|---|
| GA-EBM | 5 | 3000 | 0.1 | $195.16 \pm 18.46$ | $17.14 \pm 0.92$ |
| PREP-EBM | 10 | 1250 | 0.2 | $206.47 \pm 28.74$ | $16.10 \pm 0.48$ |
| PREP-EBM+HIP | - | - | - | $70.74 \pm 7.70$ | - |
| PREP-EBM+HIP+HT | - | - | - | $69.37 \pm 6.32$ | $19.30 \pm 1.06$ |

*3.6. Mechanical Properties*

The microhardness in the X-Z plane of GAed and PREPed samples under different conditions are compared with wrought GH4099 sample. The results show that the hardness value of the as-EBM GAed sample ~368 µm is equivalent to the wrought sample ~369 µm and is slightly lower than the hardness standard of the wrought GH4099 superalloy going through standard HT ~376 µm. The mechanical properties of GH4099 Ni-based superalloy in EBM condition have basically reached the forging standard. The microhardness of the PREPed GH4099 sample ~359 µm is slightly lower than that of wrought condition. As mentioned earlier, microcracks existed in the as-EBM PREPed specimens, and the hardness reflects the ability of the material to resist external force deformation. When measuring the microhardness at the same height of the prepared sample in a random manner, it is likely that the position of the measuring point is on or near the cracks, resulting in a decrease in the average value of the microhardness. At the same time, the standard deviation in microhardness measurement of the PREPed sample is also large, which may be due to inhomogeneous measuring points caused by the microcracks. The same case to GA sample is presumed as follows: GA powders were rapidly cooling and solidification was caused

by argon gas during the preparation process, and there were more or less porosities inside. Therefore, the gas therein cannot escape in such a short time during a rapid solidification process, or the electron beam energy input is excessive, resulting in evaporation and ejection of the feedstock powders or molten materials, thereby generating pores in the deposited metals. Similarly, the uneven measuring points nearby porosities can lead to relatively big errors.

The microhardness in the X-Z plane of PREPed samples in EBM+HIP condition and EBM+HIP+HT condition was measured, compared with a wrought GH4099 sample after a standard HT. As shown in Figure 10, the microhardness of as-built PREPed sample decreases after the HIP treatment which is attributed to the closure of defects such as microcracks by the applied HIP treatment [42]. The γ' phase of the PREPed sample grew in an unstable manner, resulting in a large amount of a larger and irregular-shaped γ' phase. The enhancement in mechanical properties of a GH4099 Ni-based superalloy is mainly due to a precipitation strengthening effect with the main precipitates of γ'. It is beneficial to improve the mechanical properties as fine γ' precipitates dispersing uniformly in the γ matrix. The strengthening effect of a coarse irregular-shaped γ' phase relative to the γ matrix will be greatly weakened, leading to the decrease of microhardness. Subsequent HT of 1473 K solid solution for 1 h + 1173 K aging for 4 h caused the γ' phase to re-dissolve into the γ matrix in a fine dispersion during aging treatment.

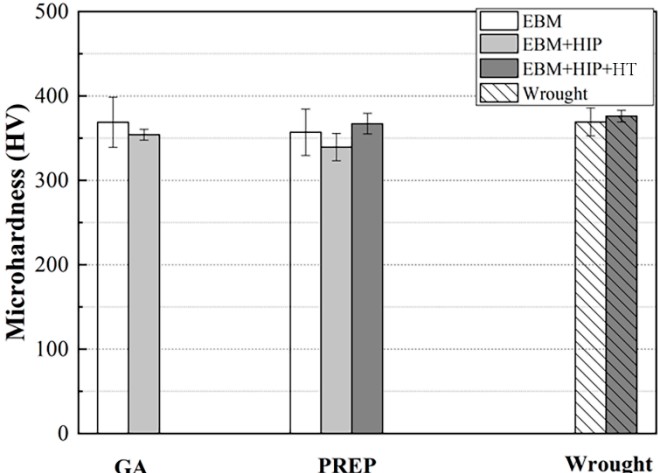

**Figure 10.** Measured average microhardness of GAed and PREPed GH4099 samples in EBM and EBM+HIP or EBM+HIP+HT condition compared with the wrought GH4099 in both as-wrought and wrought + HT conditions.

The elastic modulus and stacking fault energy between the two phases are different, which results in conformal strain and reinforcement effect on the alloy. Moreover, because of the larger specific area of the fine particles in the gamma phase, the reinforcement effect is significantly enhanced, which makes the PREP-EBM+HIP+HT specimens microscopic. Hardness has been improved. At the same time, cracks will not occur again after the solution aging treatment, so its microhardness is higher than that of the EBM state and EBM+HIP state. Meanwhile, the cracks did not re-generate inside the sample after HT, so its microhardness is notably enhanced compared with its in EBM and EBM+HIP conditions.

The as-EBM GAed sample was subjected to the same standard HT treatment. The strip tensile test specimen was cut parallel to the EBM building direction, its tensile strength at room temperature was measured, as well as in EBM+HIP+HT condition of PREPed samples which were cut parallel and perpendicular to the building direction, respectively. The above mentioned samples were compared with the tensile strength of a wrought GH4099 sample and by the standard HT at room temperature, as shown in the Figure 11.

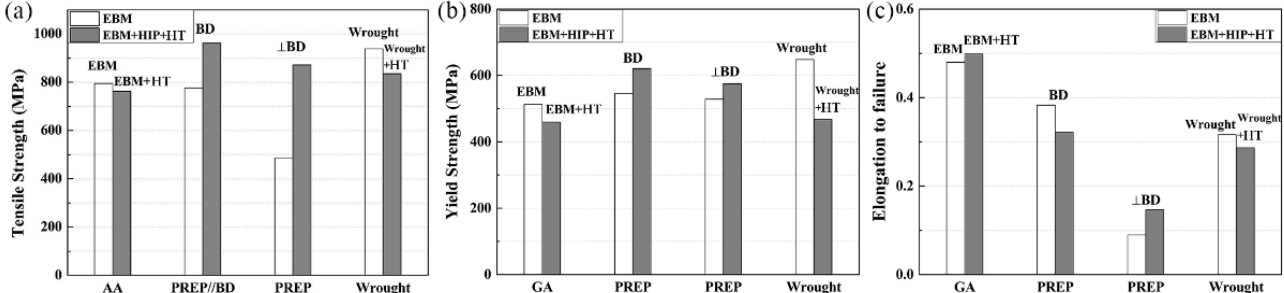

**Figure 11.** (**a**) Tensile strength, (**b**) yield strength, and (**c**) elongation to failure of GAed and PREPed GH4099 sample in different conditions cut parallel or/and perpendicular to building direction, as compared with that of a wrought sample along building direction.

Both of the tensile strengths of as-EBM GH4099 samples were lower than that of the wrought sample. Further, the tensile strength of the GAed sample is slightly higher than that of PREPed sample, which can be related to the small amount of pores inside GAed sample and the microcracks that existed in the PREPed sample. The tensile strength of post-processed PREPed sample along building direction was significantly more improved than the as-EBM sample due to the elimination of microcracks after HIP treatment and re-dissolving and re-precipitating dispersed γ′ phase during HT, which provided a high mechanical strength of about ~926 μm. Tensile strength of the PREPed sample, which was cut perpendicular to the EBM building direction (i.e., perpendicular to the growth direction of the columnar crystal) at about ~486 μm was significantly lower than that parallel to the building direction about ~775 μm, which can be related to intergranular cracks distributed in as-EBM PREPed sample along grain boundary, resulting in low intergranular bonding force. Cracks rapidly expanded while stretching along building direction and then exhibited a low tensile strength. After HIP and HT, tensile strength increased to ~757 μm and was still slightly lower than the as-EBM PREPed tensile specimen parallel to the building direction indicating the anisotropy in strength. The tensile strength of the as-EBM GAed sample decreases after undergoing HT as is also the case for the wrought GH4099 sample.

## 4. Discussion

### 4.1. Crack Mechanism

GH4099 is considered difficult to weld due to its high cracking susceptibility related to high Al + Ti contents. The intergranular cracks were found particularly at columnar grain boundaries aligned along the building direction in as-EBM condition PREPed sample shown in Figure 7c, no transverse cracks and lack of fusion phenomenon were observed, indicating that the crack is an intergranular crack. Figure 12c,d show the secondary electron images of the crack surface from a PREPed sample which was cut into a 2 mm thickness sheet in the X-Z plane and pried apart running along the crack lines. Observation of the fracture surface shows the presence of discrete liquid films in quantities, which were wetting the dendrites when cracking occurred. It is clear that the high cooling rate for the EBM process leaves a magnitude of residual stresses, further supporting the notion of susceptibility of solidification cracking in presence of liquid films. On the other hand, thermal stresses during the last stage of solidification, pulling on the liquid films results in cracking [27]. The EBM melting process is quite a complicated course with re-melting, partial re-melting, cyclic annealing, etc. [27]. It is hard to identify the existing crack propagated as solidification cracking or liquation cracking. For solidification cracking, dendrite formation inhibits the flow of the remaining liquid in the interdendritic regions, which act as crack initiation points under the effect of the stress induced by solidification. Liquation cracking is generally reported to occur in a position away from the melt pool where the material is heated rapidly to a temperature which is lower than the overall liquidus of the material [43,44]. The fracture surface of crack-free GAed sample, on the

other hand, exhibited dimple-like features as a response to a typical ductile failure shown in Figure 12b.

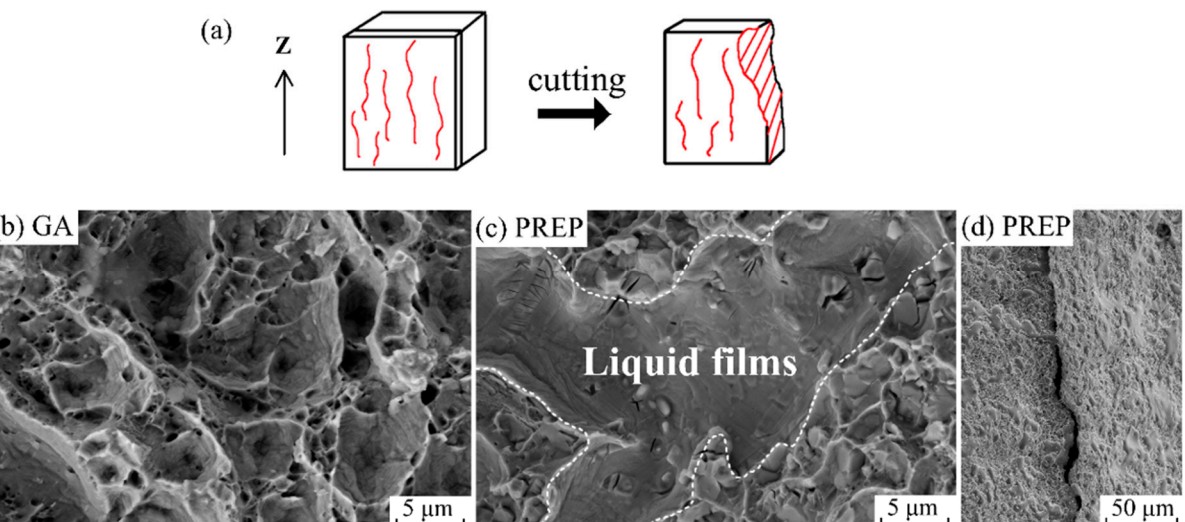

**Figure 12.** High magnification SE-SEM images of fracture surface from: (**b**) GAed sample, (**c**) a broken open PREPed sample, and (**d**) low magnification of (**c**). (**a**) Illustration of a PREPed sample that has been broken open.

Li et al. [45] reported that the crack initiated with the assistance of the transverse tensile strain/stress which tore up the liquid film formed by the low-melting point pre-existing phases in the primary heat affected zone, such as $\gamma/\gamma'$ eutectics and coarse $\gamma'$ precipitates. Thermal contraction and re-precipitation of secondary $\gamma'$ leads to a high density of dislocations at the $\gamma/\gamma'$ interface, further accelerating the separation of liquid film from grain boundary. Chauvet et al. [27] studied the fracture surface of a nonweldable Ni-based superalloy fabricated by a powder bed-based selective electron beam melting (S-EBM), and the dendritic morphology with a limited development of secondary arms indicated the presence of intergranular liquid films in the upper part. As the top layer had a limited shrinkage from former solidified layers, surface tensile effects were detrimental when tensile stress was applied to liquid film wetting dendrites. They also found that the cracks associated with the presence of liquid film propagated along high angle grain boundaries (misorientation > 15°), whereas low angle grain boundaries (misorientation < 15°) remained uncracked. Zhang et al. [46] established a mathematical model, which explained that different volume fractions of remaining liquid in CM247LC and IN792 Ni-based superalloys resulted in high strains and strain rates, thus affected crack formation and castability. The size of freezing range in the two-phase mushy zone contributes to the alloys property which decided hot tearing resistance, as per Refs. [46,47]. Moreover, hot tears occur when solid dendrite arms do not coalesce and liquid is still continuously present in between the dendrites above the coherency temperature, which is associated with deformation when tensile stresses are applied to the non-coherent dendrite network with insufficient liquid [47]. As for a nonweldable Ni-based superalloy, Kontis et al. [11] found that the amount of solutes segregation, namely of B, Mo, and Cr to HAGBs, led to the occurrence of liquid films where concentration in the liquid reached the critical composition to form borides along with the solidification stresses. They also identified hot cracking caused by grain boundary segregation, referred to as "segregation induced liquation", over the course of remelting and deposition of the subsequent layers in the additive manufacturing process.

*4.2. Effect of Si Segregation on Cracking Behaviours*

EPMA analysis represents the element distribution on fracture surface of PREPed sample in the X-Z plane. As shown in Figure 13, it can be observed that there are distinct Si

and O enrichment at the crack edge, indicating that enrichment of trace elements such as Si contributes to the crack formation.

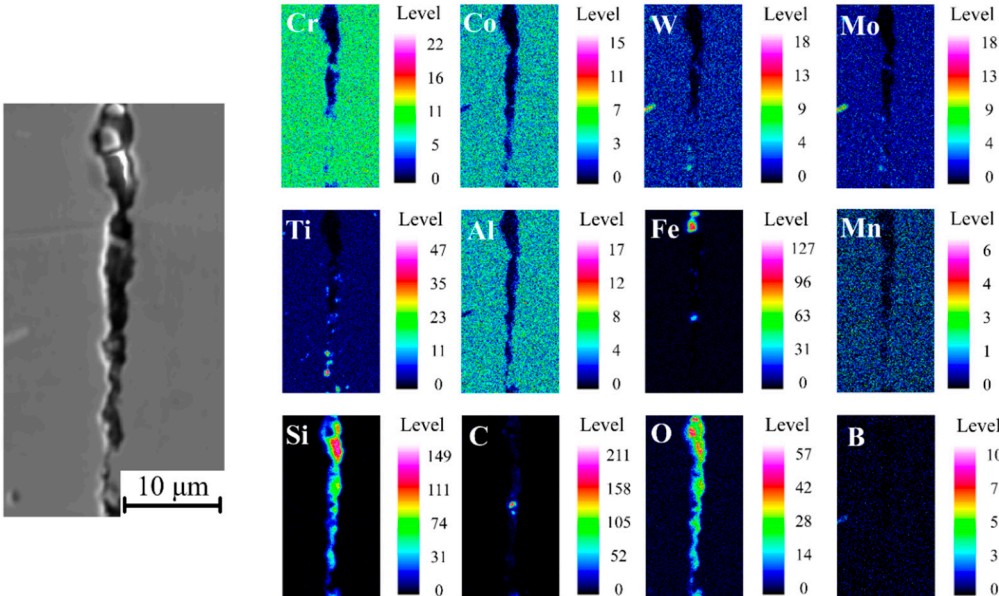

**Figure 13.** Series of elemental maps by EPMA showing the distribution near cracks of key elements in an as-EBM PREPed GH4099 superalloy. The colored scale bar at the right shows relative concentration, showing Si and O enrichment. Map conditions: 0.2 μm step size both in x and y, 20 kV.

Cloots et al. [48] revealed that cracks of IN738LC samples processed by selective laser melting propagate along building direction and distribute in a large grain boundary. Crack surfaces that seemed to be wetted with a liquid film were detected by atom probe tomography (APT), indicating the segregation of Zr at grain boundaries, which is a possible reason for hot cracking during the SLM process because it dramatically lowers the solidus temperature of IN738LC. Zhao et al. [49] investigated the fracture surface in René88DT superalloy prepared by laser solid forming (LSF) and formation of re-solidified products along grain boundaries in HAZ can be found. Energy dispersive spectroscopy (EDS) analysis results represented these re-solidified products within the cracks, containing high amount of Ti, Al, Cr, Co, and Ni, are the re-solidified ($\gamma + \gamma'$) eutectics, and speculated that HAZ cracking results from liquation cracking. Engeli et al. [50] investigated crack density of IN738LC samples with different amounts of Si content fabricated by selective laser melting and found that the fraction of Si has a strong detrimental effect to the crack density even on a small amount <0.2 wt.%, as well as its SLM processability.

It is well known from previous works that minor elements including C, B, S, P, Mg, etc. strongly influence the weldability of Ni-based superalloys even on ppm level [34]. Silicon may be used as refining addition and deoxidizer to improve weldability and oxidation resistance during traditional melting process, but its detrimental effect presented in the final alloys must be considered according to Ref. [51].

ICP-AES results of chemical composition of the GA and PREP GH4099 powders in Table 1 show that the Si content of the PREP powder is higher by one order of magnitude than that of the GA powder. It can be seen that the GH4099 sample deposited with the PREP powder had a larger content of Si from vendors mixed in, thereby increasing the crack sensitivity during the EBM forming process, inducing the occurrence of intergranular crack.

## 5. Conclusions

GH4099 superalloy samples were successfully fabricated by EBM by using GA and PREP powders. Several observations and conclusions are drawn.

(1) Dense and crack-free samples were built at 1253 K for the GA powder. By contrast, cracks were observed in the PREP sample built at 1373 K. For both cases, fine spherical $\gamma'$ phase precipitated uniformly with a volume fraction of ~20%.

(2) Local enrichment of minor element Si was responsible for the cracking behavior of the PREP sample.

(3) The cracks were successfully healed by HIP and heat treatment.

(4) Tensile properties of the GA and PREP samples in the build direction are comparable to a wrought superalloy. However, the healed cracks remain weak in the horizontal direction.

**Author Contributions:** S.W. conceived and designed the experiment, performed the experiments and wrote the paper; S.T. analyzed the data and discussed the results; H.P. designed the experiment and wrote the paper. All authors have read and agreed to the published version of the manuscript.

**Funding:** This research and the APC was funded by the National Science and Technology Major Project (J2019-VII-0016–0157) and National Key Research and Development Program of China (2021YFB3700501).

**Data Availability Statement:** Not applicable.

**Conflicts of Interest:** The authors declare no conflict of interest.

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
