# Peer review of "Influence of Powder Characteristics on the Microstructure and Mechanical Behaviour of GH4099 Superalloy Fabricated by Electron Beam Melting"

_metals, doi:10.3390/met12081301_

Round 1

Reviewer 1 Report

The paper “metals-1829909” related to electron beam powder bed fusion in AM was reviewed. Please follow the comments carefully and resubmit your paper for the next consideration and reviewing process.

  1. What is the main novelty of the paper? Please highlight it.
  2. Improve the abstract by adding short quantitative results to the abstract.
  3. Please refer to ASTM 52900 for correct terminology for AM. The correct terminology for your case is Electron Beam Powder Bed Fusion (EB-PBF).
  4. Reconsider placing fig 1 in another place.
  5. More explanation about fig 6 (EBM processing Window) is needed.
  6. How authors selected the process parameters?
  7. Please explain your design of the experiment?
  8. AM has many usages in different industries. To improve the contribution of the paper, add a short statement in the introduction by using the following papers and mention the privilege of powder bed fusion in manufacturing. “Additive manufacturing a powerful tool for the aerospace industry”
  9. Please update the introduction with the new publications in the field of laser-based powder bed fusion and compare these references with the electron beam. Authors are encouraged to read and add a short note about following new papers in the field.

·        Fatigue life optimization for 17-4Ph steel produced by selective laser melting

·        Study on the impact behaviour of arch micro-strut (ARCH) lattice structure by selective laser melting (SLM)

·        Optimization of LB-PBF process parameters to achieve best relative density and surface roughness for Ti6Al4V samples: using NSGA-II algorithm

·        Ti6Al4V scaffolds fabricated by laser powder bed fusion with hybrid volumetric energy density

Author Response

Response to Reviewer 1 Comments

Dear Editor of Metals

We would like to express our sincere thank you for your time and efforts to help us to improve the paper quality and enhance the scientific impacts. For reviewer’s comments, we have provided the point-by-point response by first providing our explanations and second summarizing what has been modified in the revised manuscript.

Point 1: What is the main novelty of the paper? Please highlight it.

Response 1: Highlights of the paper

  • EBM processing window was established and systematically compared for GA and PREP superalloy powders.
  • The influence of raw powder characteristics on the microstructure of the EBMed superalloy was investigated.
  • The effect of minor element Si on the cracking behaviour of EBM superalloy was elucidated.

Point 2: Improve the abstract by adding short quantitative results to the abstract.

Response 2: Quantitative results were added in the revised abstract (Page 1 Lines 22-31)

Point 3: Please refer to ASTM 52900 for correct terminology for AM. The correct terminology for your case is Electron Beam Powder Bed Fusion (EB-PBF).

Response 3: We checked the whole article and corrected the terminologies according to ASTM 52900. In terms of utilization in the aerospace industry, the powder bed fusion (PBF) techniques comprise of selective laser melting (SLM) and electron beam melting (EBM). In this work, the terminology electron beam melting (EBM) was used.

Point 4: Reconsider placing fig 1 in another place.

Response 4: According to the reviewer’s comment, Fig. 1 was replaced with reference citations. See Refs [12, 13, 24].

Point 5: More explanation about fig 5 (EBM processing Window) is needed.

Response 5: More explanations about the EBM processing window were added. Please find on Pages 7 Lines 217-222.

Point 6: How authors selected the process parameters?

Point 7: Please explain your design of the experiment?

Response 6&7: Firstly, building temperature was confirmed by performing powder sintering tests. Optimized temperature of 1253 K and 1373 K was selected for GA and PREP powders in the following additive manufacturing. The slightly sintered powder bed was necessary for the process stability of preheating and melting during EBM.

Point 8: AM has many usages in different industries. To improve the contribution of the paper, add a short statement in the introduction by using the following papers and mention the privilege of powder bed fusion in manufacturing. “Additive manufacturing a powerful tool for the aerospace industry”

Response 8: The authors are very grateful to the reviewer for the suggestion of adding some usages in different industries by AM process in Introduction section. We clarified the privilege of powder bed fusion in manufacturing, especially the view point of AM is a powerful tool for the aerospace industry by using some latest published references, which can be found in the revised manuscript on Page 1-2 Lines 36-47.

Point 9: Please update the introduction with the new publications in the field of laser-based powder bed fusion and compare these references with the electron beam. Authors are encouraged to read and add a short note about following new papers in the field.

Response 9: References [1-7] were updated in the introduction section according to the reviewer’s comment.

Reviewer 2 Report

The authors have presented an interesting topic - additive manufacturing of Ni superalloys. They have presented a comparion of two types of metallic powders and have shown their differences advantages and disadvantages. They have described their findings in detail and have provided convincing evidence of the issues they encountered, like cracking of the PREP powder and the closing of the cracks by using HIP, but still exposing the potential weakness and that these imperfections cannot be remedied completely, as the material still shows some weakness in the horizontal direction (perpendicular to the building direction). I have only a few minor issues I wish to address, but overall I think this is a good paper that will spark interest in readers.

-issue 1: The abbreviation STA is used for the heat treatment, please do not use it, as STA is commonly used for simultaneous thermal analysis in materials science.

-issue 3: The heat treatment experiment, please add, whether you used an air atmosphere furnace, were the samples put in a hot or cold furnace, how was the temperature controlled (just furnace or thermocouple in a sample), how did you detect that the samples reached the temperature, and when did the 1 h count down begin, and the same for ageing

-issue 4: Figure 9, please describe a) and b) in the figure captions

-minor language mistakes, like precipitates do not grow up, they just grow, and binding defects is used once instead of bonding

Author Response

Response to Reviewer 2 Comments

Dear Editor of Metals

We would like to express our sincere thank you for your time and efforts to help us to improve the paper quality and enhance the scientific impacts. For reviewer’s comments, we have provided the point-by-point response by first providing our explanations and second summarizing what has been modified in the revised manuscript.

Point 1: The abbreviation STA is used for the heat treatment, please do not use it, as STA is commonly used for simultaneous thermal analysis in materials science.

Response 1: The authors are very grateful to the reviewer’s kind suggestion. Authors have checked the whole manuscript and replace the abbreviation STA (solution treatment and aging) with HT (heat treatment).

Point 2: The heat treatment experiment, please add, whether you used an air atmosphere furnace, were the samples put in a hot or cold furnace, how was the temperature controlled (just furnace or thermocouple in a sample), how did you detect that the samples reached the temperature, and when did the 1 h count down begin, and the same for ageing.

Response 2: According to the reviewer’s comments, heat treatment details were provided in the updated manuscript. Please find on Page 6 Lines 159-165.

Point 3: Figure 8, please describe a) and b) in the figure captions.

Response 3: Caption of Fig.8 was revised (Page 10 Lines 293-294).

Point 4: minor language mistakes, like precipitates do not grow up, they just grow, and binding defects is used once instead of bonding.

Response 4: The language of the manuscript was carefully polished. Mistakes were corrected in the updated version.

Round 2

Reviewer 1 Report

The paper is ready to publish.